# Habitual Mask Wearing as Part of COVID-19 Control in Japan: An Assessment Using the Self-Report Habit Index

**DOI:** 10.3390/bs13110951

**Published:** 2023-11-19

**Authors:** Tianwen Li, Marie Fujimoto, Katsuma Hayashi, Asami Anzai, Hiroshi Nishiura

**Affiliations:** Graduate School of Medicine, Kyoto University, Yoshida-Konoe-cho, Sakyo-ku, Kyoto 606-8501, Japan; li.tianwen.44e@st.kyoto-u.ac.jp (T.L.); fujimoto.marie.86w@st.kyoto-u.ac.jp (M.F.); hayashi.katsuma.7w@kyoto-u.ac.jp (K.H.); anzai.asami.43c@st.kyoto-u.ac.jp (A.A.)

**Keywords:** cultural tightness, behavior, precaution, social norms, cross sectional study, epidemiology

## Abstract

Although the Japanese government removed mask-wearing requirements in 2023, relatively high rates of mask wearing have continued in Japan. We aimed to assess psychological reasons and the strength of habitual mask wearing in Japan. An Internet-based cross-sectional survey was conducted with non-random participant recruitment. We explored the frequency of mask usage, investigating psychological reasons for wearing masks. A regression analysis examined the association between psychological reasons and the frequency of mask wearing. The habitual use of masks was assessed in the participant’s most frequently visited indoor space and public transport using the self-report habit index. The principal component analysis with varimax rotation revealed distinct habitual characteristics. Among the 2640 participants surveyed from 6 to 9 February 2023, only 4.9% reported not wearing masks at all. Conformity to social norms was the most important reason for masks. Participants exhibited a slightly higher degree of habituation towards mask wearing on public transport compared to indoor spaces. The mask-wearing rate was higher in females than in males, and no significant difference was identified by age group. Daily mask wearing in indoor spaces was characterized by two traits (automaticity and behavioral frequency). A high mask-wearing frequency has been maintained in Japan during the social reopening transition period. Mask wearing has become a part of daily habit, especially on public transport, largely driven by automatic and frequent practice.

## 1. Introduction

The coronavirus disease 2019 (COVID-19) is a respiratory disease caused by severe acute respiratory syndrome coronavirus 2 (SARS-CoV-2), which has resulted in over 680 million confirmed deaths as of April 2023 [1,2]. The virus is transmitted mainly via droplets and aerosols [1,3,4,5,6]. Secondary transmission is known to occur even during the incubation period (i.e., pre-symptomatic stage), accounting for approximately half of all secondary transmissions [7], and transmission from asymptomatic or mildly infected individuals can also occur [8]. To control the spread of COVID-19, a concerted effort that combines both pharmaceutical and non-pharmaceutical interventions was carried out across the world. While none of those interventions perfectly prevented transmission or severe illness, as part of a suppression policy to curb epidemic waves and slow transmission rates, many nations across the world implemented a range of public health and social measures (PHSMs), especially before the completion of the primary series vaccination [9]. The PHSMs included but were not limited to the restriction of contact (including lockdowns), contact tracing, social distancing, and personal protections, including mask-wearing [9,10,11,12]. While the vaccines contributed to greatly reducing the disease burden [13], especially by elevating the indirect protection of the vaccinated population, the fatality risk among vulnerable people remained high [14]. Moreover, post-COVID-19 sequelae are highly prevalent [15,16].

Although many of the PHSMs that greatly reduced contact were temporarily effective in reducing the incidence [17], the PHSMs can lead to pandemic fatigue, resulting in diminished effectiveness and responsiveness among citizens [18,19]. Ideally, sustainable countermeasures would be able to control the epidemic for an extended time period without causing fatigue. While the individual benefit of wearing a mask in protecting oneself is yet debated, mask wearing has been considered a cost-effective preventive measure [20]. During the pandemic, mask wearing has sometimes been recommended or mandated for entire populations, because secondary transmission from asymptomatic infected individuals can occur without recognition [6,21,22,23]. The wearing of non-woven fabric masks by infected individuals contributed to a reduction in the risk of secondary transmission [8]. A systematic review of 21 papers on this subject found that community-wide recommendations to wear masks were associated with reduced numbers of new infections, hospital admissions, and deaths [22]. While compliance with mask wearing is a potential practical issue, mask wearing is considered to be more sustainable than other non-pharmaceutical interventions, such as staying at home or maintaining physical distance [18]. Many countries have promoted de-masking as a policy symbol of downgrading COVID-19 control and fully reopening society [18,24,25]. Japan has not been an exception, and the government announced on 13 March 2023 that people may actively decide for themselves whether to wear masks, even in indoor spaces [26].

Given the circumstances described above, it is important to understand the psychological reasons and extent of the habituation of mask wearing. Published studies have frequently examined the reasons why people wear masks using various methodological approaches [27,28,29,30,31,32]. Nakayachi et al. suggested that in Japan, mask wearing in the early stages of the pandemic was driven by peer pressure rather than personal choice [30]. Little is known about the potential for habituation of mask wearing, even three years after the onset of this pandemic. Wood et al. conducted a literature review and attempted to propose several mechanisms of habituation [33], concluding that nonhabitual behaviors tend to induce greater levels of stress, whereas habitual behaviors employed in daily tasks reduced stress and increased the likelihood of success in continuing the behavior [34].

Do the Japanese still wear the mask due to peer pressure and to what extent is the mask-wearing behavior already habitual among the Japanese? The purposes of the present study were to understand the mask-wearing rate and its psychological reasons and also to measure the strength of mask-wearing habits among Japanese adults. To achieve the second objective, we used the self-report habit index (SRHI), originally developed by Verplanken and Orbell to assess habit strength [35], and applied it to mask wearing in Japan. The SRHI has been widely used across many health fields [36] and translated into Japanese, and its reliability has been confirmed elsewhere [37]. The SRHI assesses three aspects of habit: automaticity, frequency of the behavior, and self-identity [35,36,38].

## 2. Materials and Methods

### 2.1. Data Source

A snapshot survey of Japanese adults aged over 20 years old was carried out from 6 to 9 February 2023. The participants were non-randomly recruited from registered users of Mellinks Ltd., Tokyo, Japan, a Japanese internet search company. The respondents did not receive financial remuneration, but upon completing the survey, they earned points from the company that could be later redeemed for valuable goods. Area sampling was performed, proportionally recruiting participant samples according to prefectural population size, with even weighting for age group (20–29, 30–39, 40–49, 50–59, 60–69, and over 70 years) and sex (female and male). While sampling people in different geographic areas and ages, the recruited people are restricted to those who can manage their time to correspond to the survey request.

### 2.2. Data Collection and Definitions

The self-reported internet survey consisted of five sections, as follows: (i) demographic characteristics (age, sex, marital status, parental status), (ii) frequency of mask usage, (iii) the psychological reasons for mask wearing, (iv) self-recognition associated with habitual use of masks at the participant’s most frequently visited indoor space other than their household, (v) and self-recognition associated with habitual use of masks on public transport. In the second section, the frequency of mask wearing in indoor spaces (other than household) was examined, using a three-point scale (1 = I have not worn one at all, 2 = I have sometimes worn one, and 3 = I have usually worn one). In the third section, referring to a published study in early 2020 [30], we explored six psychological reasons for wearing masks. That is, we explored six psychological items that represent possible reasons to wear a mask, as follows: (i) perceived severity of disease (severity), (ii) wearing a mask to protect oneself (protection), (iii) wearing a mask to prevent spread (prevention) (iv), conforming to a norm to wear masks (norm), (v) feeling relief when mask wearing (relief), and (vi) impulse to take whatever actions are necessary (impulsion), via a five-point Likert scale (1 = not at all to 5 = very much).

The fourth section gathered information about the respondents’ most frequently visited indoor spaces and the strength of mask-wearing habits among respondents in that particular indoor space. To measure the strength of habit, we used the Japanese version of the SRHI [37] using a seven-point Likert scale (1 = agree to 7 = disagree). The fifth section gathered information on the most frequently used public transport and the strength of mask-wearing habits on public transport. Again, we used the Japanese version of the SRHI. English-translated survey questions are available as a Appendix A.

### 2.3. Statistical Analysis

Statistical analysis was carried out in two steps. In the first step, we explored the frequency of mask wearing among participants and explored its relationship with psychological reasons. Subsequently, we investigated the strength of mask-wearing habits. Sex, marital status, parental status, age groups, most frequently visited indoor space, and most frequently used public transport were dealt with as categorical variables, and all other variables were dealt with as continuous variables. The level of significance was set at α = 0.05.

To summarize the results, we first clarified the descriptive features of the frequency of mask wearing (not at all, sometimes, and usually). To explore differences in the mask-wearing rate by demographic variables, multinomial logistic regression was used. Multiple linear regression was then used to model the relationship between the frequency of mask wearing and the six psychological reasons. The 12 items of the SRHI were assessed via principal component analysis (PCA) using maximum likelihood extraction and varimax rotation. The item loadings were derived from the pattern matrix. Subsequently, the SRHI item frequencies were compared to those in component 1 (the first principal component) and those in component 2 (the second principal component). All statistical data were analyzed using JMP Pro statistical software, version 17.0 (SAS Institute Inc., Cary, NC, USA).

### 2.4. Ethical Considerations

The participants received an explanation about the background and purpose of the study before enrollment and were given an explicit right to withdraw from the study. They were then asked to review the consent document prior to responding to our survey, and only those who agreed via the webpage were included. After the survey, Mellinks compiled anonymously processed information that could not be linked to any personally identifying information. This study was approved by the Medical Ethics Board of the Graduate School of Medicine at Kyoto University (R3863).

## 3. Results

From 6 to 9 February 2023, a total of 2640 participants responded to our survey. The demographic characteristics of participants are summarized in Table 1. Sex and age were evenly distributed by sampling, and 45.5% of participants were unmarried. Offices (*n* = 1305; 49.4%) were the most frequently visited indoor spaces other than households, followed by shopping malls (n = 935; 35.4%) and healthcare facilities (n = 170; 6.4%). Trains (n = 1709; 64.7%) were the most frequently used public transport, followed by buses (n = 499; 18.9%).

Table 2 shows the frequency of mask wearing. Overall, 83.7% of the participants reported that they usually wore a mask, and only 4.9% reported that they did not wear a mask at all. Female participants reported wearing masks more frequently than male participants (88.0% vs. 79.4% of female and male participants, respectively, responded “usually”), but no clear differences were identified by marital status. We did not identify significant differences in mask-wearing rates by age group.

Exploring psychological reasons for wearing masks, all six psychological reasons were positively correlated with the frequency of mask usage (Appendix A), which is consistent with previous reports [27,30]. Table 3 shows the associations between the frequency of mask wearing and six psychological reasons. According to the standardized coefficient, conformity to mask-wearing norms (0.22, *p* < 0.0001) was the most important reported psychological reason and was the only reason exhibiting a statistically significant association with the frequency of mask wearing. This association was also validated in different subgroups (Appendix A).

Figure 1 shows the distribution of the SRHI scores in indoor spaces and on public transport, stratified by sex and age group. The total SRHI score of all the participants in indoor spaces was 64.9 (±15.7). When stratified by sex, female participants scored 68.2 (±14.2) and male participants scored only 61.5 (±16.5). Additionally, older participants (aged 65 years and older) scored 70.0 (±14.0), while younger participants were less habituated to mask wearing, with a score of 63.1 (±15.9). The total SRHI score on public transport was 67.2 (±18.4), which was greater than that in indoor space. Stratifying by sex, female participants scored 70.8 (±16.1) and male participants scored 65.6 (±19.8). Additionally, older participants scored 71.3 (±18.8), while younger participants scored 65.8 (±18.0). In most subgroups, the SRHI scores on public transport were higher than those in indoor spaces (Appendix A).

Table 4 shows the results of PCA with varimax rotation of SRHI. For mask wearing in indoor space, the 12 items were underpinned by two components (component 1: eigenvalue 7.9, 66.1% variance explained, and component 2: eigenvalue 1.0, 8.1% variance explained). Component 1 captured 10 items, while component 2 captured nine items. The 12 items related to mask wearing on public transport were grouped under one component (component 1), which had an eigenvalue of 9.2 and explained 76.7% of the variance. Stratified by demographic characteristics, the results of the PCA of the SRHI were consistent in most subgroups (Appendix A). For both indoor spaces and public transport, items 2, 3, 5, and 8 are known to reflect the automaticity of mask wearing behavior, yielding higher coefficient values than other items. Moreover, items 4, 6, 9, and 11 reflect self-identity, and small coefficient values for this category indicate that mask wearing had little to do with self-identification.

## 4. Discussion

Japan is considered to have a higher level of cultural tightness compared with Western countries [39]. We conducted a snapshot survey to investigate the prevalence of mask wearing in February 2023 in the Japanese cultural context. Of the 2640 participants, 83.7% reported that they usually wear a mask, and the behavior was found to be regulated mainly by social norms. Additionally, we assessed whether mask wearing had become a daily habit and found that respondents wear masks more regularly and unconsciously on public transport compared with indoor spaces. A daily habit of mask wearing was more likely to be reported by women and older people (65 years and older). PCA revealed that items associated with automaticity of behavior yielded high coefficient values, while mask wearing had little to do with self-identity.

Even shortly before the government legally downgraded the category of COVID-19 to be comparable to that of seasonal influenza from May 2023, mask wearing was maintained at a very high level in Japan, exceeding that at the early stages of the pandemic in 2020 [30]. Consistent with previous studies [29,40,41], female participants reported wearing masks more frequently than male participants and exhibited more adherence to personal protective behaviors, whereas some male respondents reported feeling ashamed of regular mask wearing [41,42,43]. COVID-19 was reported to change personal protective behaviors, and the rate of mask wearing did not differ by age. Positive emotional appraisals of masks were previously reported to be associated with the frequency and duration of mask wearing [44], the interpretation of which is linked to the theory of micro-valences [45]. From 2020 to 2023, mask wearing in Japan remained a strong social norm, and because of the emotional value of social norms, people in Japan were more likely to wear masks. Moreover, the collectivism that stems from social harmony and cohesiveness helped to positively promote mask wearing during the COVID-19 pandemic [28,41].

According to our participants, conformity to social norms was the most important reason for mask wearing, and the involvement of peer pressure has not changed since early 2020 [30]. Our finding regarding this point is consistent with the results of another study [27], which revealed that wearing masks was what people tended to express serious concern about non-mask-wearing behavior in the Japanese cultural context. This phenomenon is thought to be partially attributable to social anxiety toward COVID-19, and the fundamental need to be accepted by others [46]. Engagement in mask wearing appears to be affected by social judgments from others [47,48]. One previous study [46] referred to this phenomenon as a “situational norm”.

After spending more than 3 years under a state of emergency during the pandemic, mask-wearing behavior became a daily habit for many people, as assessed by SRHI items [35]. We found that people exhibited stronger habituation to mask wearing on public transport compared with indoor places. The need to wear masks on public transport is in line with social norms, and the finding is also partially explainable by ecological psychology theory [49,50], in which people’s behaviors and social thoughts are thought to be determined by time, physical environment, and the duration of time spent in a particular environment. That is, during weekdays, people spend longer time in indoor spaces than on public transport, and thus, may be more familiar and feel safer with indoor spaces, leading them to wear masks less frequently. Women have been found to wear masks more frequently than men, have more positive attitudes regarding their effectiveness, and perceive a higher level of risk than men [29,40,41]. All these factors may have contributed to a stronger habitual tendency for mask wearing in women compared with men. A previous study [51] reported that older people wore masks more often to avoid serious risks, possibly because they are more vulnerable to severe complications. In the long term, older people are likely to continue to habitually wear masks. Ongoing habituation in the presence of social norms is of course preferable news for public health, but not necessarily good news to the future society with downgraded COVID-19 control. Even provided that the risk of infection is lowered at some point in the future time, there might be strong social norms on public transport, and people who clearly do not wish to wear the mask need to maintain a non-habitual behavior of wearing the mask even in the society with reduced risk of infection.

Our PCA results indicated a one-dimensional structure, suggesting that the Japanese version of the SRHI can be used to assess the strength of habitual mask usage. We found that two characteristics underlie the habituation of wearing masks in indoor spaces, but only one characteristic applies to public transport. Based on previous studies [35,36,38], we interpreted the two rotated components as capturing the different characteristics of habit: automaticity, and frequency of the behavior, separately. When people spend time in public spaces, their behaviors are influenced by social norms, and 3 years of practice during the pandemic led to mask-wearing behavior being exercised unconsciously. With strong social norms, people tend to feel safer mimicking the behaviors of others and obeying guidance from others, so their social behavioral effectiveness can be maximized [46]. However, according to ecological psychology [50], we regard that people who spend most of their time in indoor spaces feel more at ease and less constrained and are more likely to realize that the behavior suits them and that this feeling allows them to be diverse and flexible in their behaviors, which leads to various characteristics of their behavior. As an important policy implication, governmental organizations should indicate specific indoor settings where habitual mask-wearing behavior is recommended in the long run (and where such indication can be relaxed).

The current study involved several limitations. First, the present study was cross-sectional, and we were not able to clarify the causal associations between mask wearing and dynamic features of collective behavior. Second, our participants were recruited using convenience sampling. This method may have led to bias in the sampling frequency. For example, individuals who receive care in a nursing home or hospital may have been less likely to participate in our study and may wear masks more frequently than the individuals who responded. Due to the convenient nature of sampling, those who do not regularly wear the mask might have been less likely to be recruited, and thus, we might have potentially overestimated mask-wearing coverage. Third, the detailed characteristics of the principal components we identified are still controversial, and the survey results regarding daily habitual behavior should be explored longitudinally. We plan to implement serial cross-sectional surveys in the future to address this issue. Fourth, the strength of habitual mask usage was assessed using a self-report method. Because individuals’ behaviors dynamically change, it will be important to consistently assess the strength of habitual mask use in future studies.

Despite these limitations, the current results demonstrate that the frequency of mask wearing remained very high in Japan in February 2023, and that this can be partially explained by the existence of strong social norms. Public transport was shown to have the potential to make mask wearing part of daily habit, while habitual use of masks was less common in the most frequently used indoor space. These findings may be helpful for informing future personal protective behaviors during the ongoing health impacts of endemic COVID-19.

## 5. Conclusions

In this study, we conducted an Internet-based cross-sectional survey between 6–9 February 2023. We revealed that a high mask-wearing frequency has been maintained in Japan during the social reopening transition period. Using regression analysis, we also found the main psychological reason for wearing masks is still conformity to social norms. And mask wearing has become a part of daily habits, especially on public transport, largely because of automaticity and behavioral frequency.

## Figures and Tables

**Figure 1 behavsci-13-00951-f001:**
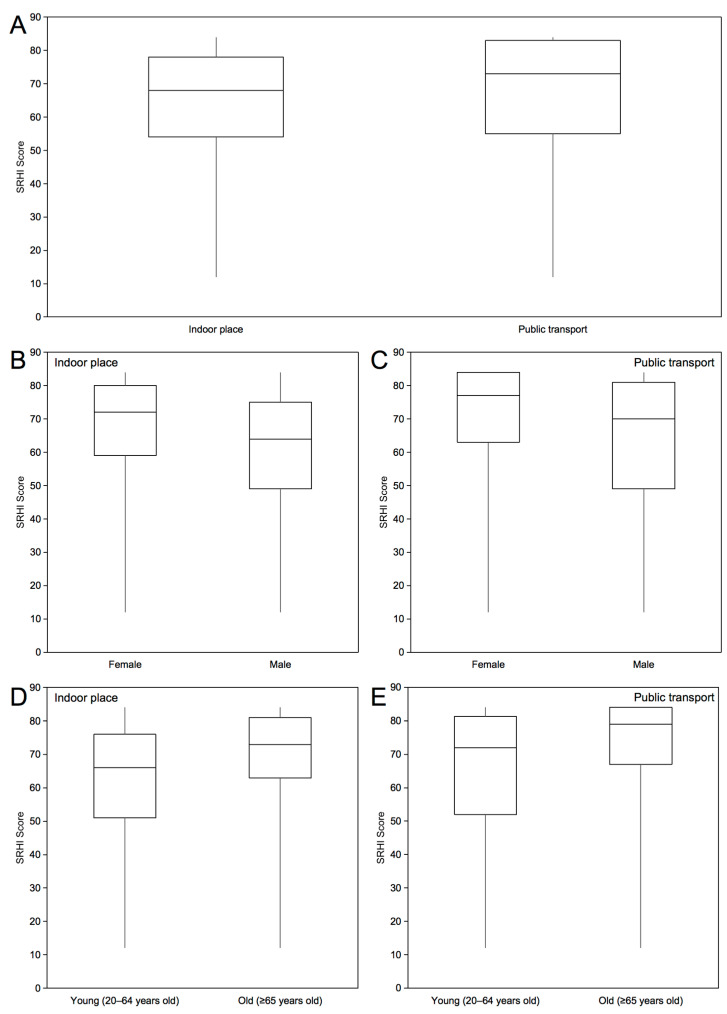
Distribution of Self-Report Habit Index (SRHI) scores. This figure shows the distribution of SRHI scores of different subgroups in indoor space and public transport. The horizontal axis represents the different subgroups, and the vertical axis represents the SRHI score. The SRHI score represents the strength of the habit of mask wearing, ranging from 12 to 84. Higher SRHI scores indicate stronger habits. (**A**) Distribution of SRHI scores of all participants in indoor space and on public transport; (**B**,**C**) Distribution of the SRHI scores by sex in indoor space and on public transport; (**D**,**E**) Distribution of SRHI scores by age group in indoor space and on public transport. The box plot ranges from lower to upper quartiles, and the middle line represents the median value. Whiskers extend to minimum and maximum scores.

**Table 1 behavsci-13-00951-t001:** Characteristics of participants (n = 2640).

Characteristics		Estimate ^†^
Sex	Male	1320 (50.0)
Marital Status	Unmarried	1200 (45.5)
Have at least one child or not	Yes	1247 (47.2)
Age group (years, all age groups have the same sample size of 440)	20–29	24.7 ± 2.8
	30–39	34.6 ± 2.9
	40–49	44.5 ± 2.9
	50–59	54.5 ± 2.9
	60–69	64.6 ± 2.9
	≥70	74.7 ± 4.0
Indoor space that is most frequently visited	Office	1305 (49.4)
	School	98 (3.7)
	Healthcare Facility	170 (6.4)
	Shopping Mall	935 (35.4)
	Public Facility	91 (3.5)
	Other Indoor Places	41 (1.6)
Public transportation is most frequently used	Train	1709 (64.7)
	Bus	499 (18.9)
	Aircraft	47 (1.8)
	Boat	9 (0.3)
	Other Public Transportation	376 (14.2)

^†^ Mean ± standard deviation; n (%).

**Table 2 behavsci-13-00951-t002:** Distribution of frequency of mask usage (n = 2640).

		Frequency of Mask Wearing		
		Not at all	Sometimes	Usually
		n (%)	n (%)	n (%)
Total		130 (4.9)	301 (11.4)	2209 (83.7)
Sex *	Male	82 (6.2)	190 (14.4)	1048 (79.4)
	Female	48 (3.6)	111 (8.4)	1161 (88.0)
Marital Status	Married	72 (6.0)	134 (11.2)	994 (82.8)
	Unmarried	58 (4.0)	167 (11.6)	1215 (84.4)
Have at least one child or not **	Yes	51 (4.1)	156 (12.5)	1040 (83.4)
	No	79 (5.7)	145 (10.4)	1169 (83.9)
Age (years)	20–29	29 (6.6)	40 (9.1)	371 (84.3)
	30–39	20 (4.6)	43 (9.8)	377 (85.7)
	40–49	27 (6.1)	53 (12.0)	360 (81.8)
	50–59	20 (4.6)	54 (12.3)	366 (83.2)
	60–69	18 (4.1)	44 (10.0)	378 (85.9)
	≥70	16 (3.6)	67 (15.2)	357 (81.1)

* *p* < 0.0001, ** *p* = 0.05. The test level is α = 0.05; the *p*-value was calculated by multinomial logistic regression analysis.

**Table 3 behavsci-13-00951-t003:** Association between mask wearing and psychological reasons (n = 2640).

Variable	Standardized Coefficient	*p*-Value
Severity	0.02 (0.02)	0.41
Protection	0.04 (0.03)	0.15
Prevention	0.02 (0.02)	0.36
Impulsion	0.05 (0.03)	0.05
Norm	0.22 (0.03)	<0.0001
Relief	0.01 (0.03)	0.59

Frequency of mask usage as explained by six variables. Dependent variable: the frequency of mask wearing; independent variable: six psychological reasons variables. Standard errors are shown in brackets. R^2^ = 0.10, R^2^(adj) = 0.09, root mean square error (RMSE) = 0.49. The test level is α = 0.05; the *p*-value is calculated by multiple linear regression.

**Table 4 behavsci-13-00951-t004:** Principal component analysis of Self-Report Habit Index (n = 2640).

	Mask Wearing Indoors				Mask Wearing on Public Transport
	Before Rotation		After Rotation		
	Component 1	Component 2	Component 1	Component 2	Component 1
Eigenvalue	7.9	1.0	4.8	3.4	9.2
Variance explained (%)	66.07	8.05	40.30	28.34	76.70
Item loadings					
1. I do frequently.	0.82	−0.32	0.78	-	0.87
2. I do automatically.	0.85	−0.22	0.80	-	0.91
3. I do without having to consciously remember.	0.88	−0.23	0.84	-	0.93
4. that makes me feel weird if I do not do it.	0.82	0.29	0.47	0.70	0.88
5. I do without thinking.	0.88	−0.09	0.74	0.47	0.93
6. that would require effort not to do it.	0.61	0.60	-	0.62	0.74
7. that belongs to my (daily, weekly, monthly) routine.	0.84	−0.22	0.72	0.41	0.90
8. I start doing before I realize I’m doing it.	0.81	−0.04	0.62	0.48	0.86
9. I would find hard not to do.	0.84	0.26	0.46	0.75	0.90
10. I have no need to think about doing.	0.79	−0.05	0.59	0.48	0.87
11. that’s typically “me”.	0.75	0.41	-	0.75	0.80
12. I have been doing for a long time.	0.85	−0.16	0.69	0.46	0.89

Component 1: the first principal component; Component 2: the second principal component; Rotation: maximum likelihood extraction and varimax rotation method. Rotated component loading values < 0.4 were not reported.

## Data Availability

The datasets used during the current study are available from the corresponding author upon reasonable request.

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
