# Peer review of "Habitual Mask Wearing as Part of COVID-19 Control in Japan: An Assessment Using the Self-Report Habit Index"

_behavsci, 2023, doi:10.3390/bs13110951_

Round 1
Reviewer 1 Report
Comments and Suggestions for Authors
The study is interesting, revealing the reality of Japan in relation to the use of respiratory protection after 4 years of the covid pandemic beginning.
It has details to improve:
Abstract: Lines 18-19 the sentence is not relevant, the grouping of components in the resume is difficult to understand. It would be of interest to describe results as the greatest use in women and older people.
Statistical analysis: please explain with more detail the use of “components” that are shown later in Table 4.
Results: line 137-138: Better to use % than raw numbers.
Table 1: Instead of using * in the bottom of the table, please insert the explanation “all age groups have the same sample size of 440” between brackets in the table to avoid confusion.
Lines 150-152 should be in methods.
Figure 1 is complicated to interpret. Normally a figure should be self-explanatory. Perhaps consider showing the results in a box plot, which in turn integrates and compares the scores between sex and age groups in the given contexts (indoor/transport). This would also reduce the size and quantity of sub-figures in the figure and improve its visualization.
At the bottom of the figure (description), C-F shows distribution by sex and context; G-J shows distribution by age and context.
Author Response
[Point-by-point responses to review comments]
Manuscript ID: behavsci-2683978. Title: Habitual mask wearing as part of COVID-19 control in Japan: An assessment using the self-report habit index. Authors: Tianwen Li, Marie Fujimoto, Katsuma Hayashi, Asami Anzai, Hiroshi Nishiura
[Responses to the Reviewer #1]
The study is interesting, revealing the reality of Japan in relation to the use of respiratory protection after 4 years of the covid pandemic beginning. It has details to improve:
- Abstract: Lines 18-19 the sentence is not relevant, the grouping of components in the resume is difficult to understand. It would be of interest to describe results as the greatest use in women and older people.
>>
We thank the reviewer for this comment. We have modified the sentence that was mentioned to be irrelevant, and moreover, we additionally added descriptions about the prevalence of mask usage in women and older people (P1L19-22).
- Statistical analysis: please explain with more detail the use of “components” that are shown later in Table 4.
>>
We additionally described what we compared by components in Table 4 (P3L136-138).
- Results: line 137-138: Better to use % than raw numbers
>>
We agree and we decided to show both raw numbers and % (P4L153-156).
- Table 1: Instead of using * in the bottom of the table, please insert the explanation “all age groups have the same sample size of 440” between brackets in the table to avoid confusion.
>>
Corrected accordingly (Table 1, P4)
- Lines 150-152 should be in methods.
>>
We agree and we moved the sentence to Methods (P3L07-108).
- Figure 1 is complicated to interpret. Normally a figure should be self-explanatory. Perhaps consider showing the results in a box plot, which in turn integrates and compares the scores between sex and age groups in the given contexts (indoor/transport). This would also reduce the size and quantity of sub-figures in the figure and improve its visualization. At the bottom of the figure (description), C-F shows distribution by sex and context; G-J shows distribution by age and context.
>>
We thank the reviewer for this suggestion. Figure 1 was revised (P6).
Reviewer 2 Report
Comments and Suggestions for Authors
Review for the article "Habitual Mask Wearing as Part of COVID-19 Control in Japan: An Assessment Using the Self-Report Habit Index"
The study under consideration, titled "Habitual Mask Wearing as Part of COVID-19 Control in Japan: An Assessment Using the Self-Report Habit Index," presents valuable insights into the continued practice of mask-wearing in Japan even after the removal of government-mandated requirements in 2023.
Introduction: The introduction of the article effectively sets the stage by addressing the continued high rates of mask-wearing in Japan, even after official mandates were lifted. However, the introduction could be enhanced by explicitly stating the research questions and objectives of the study. By doing so, the authors could provide a clear roadmap for readers, guiding them on what to expect in the study.
Methodology: The methodology section of the study is commendable for its clarity and detail. The use of an Internet-based cross-sectional survey is well-suited to capture a large sample of participants. However, it would be beneficial to further elaborate on the recruitment process and discuss any potential limitations associated with non-random participant recruitment.
Results and Findings: The article reports that only 4.9% of the 2,640 participants surveyed reported not wearing masks at all. It highlights the importance of conformity to social norms as a primary reason for mask-wearing, and it underscores the habituation towards mask-wearing, particularly in public transport. This section presents a clear picture of the study's key findings.
Discussion: While the discussion section effectively links the findings to the existing literature, it could be strengthened by a more comprehensive exploration of the implications of these findings. For instance, how might this ongoing habituation to mask-wearing impact public health and social norms in Japan? Additionally, the article could discuss potential policy implications and recommendations based on the results.
Conclusion: The conclusion succinctly summarizes the key takeaways from the study, emphasizing that mask-wearing in Japan has become a daily habit, particularly in public transport, due to automatic and frequent practice.
Overall Appreciation: This study is a valuable contribution to understanding the persistent practice of mask-wearing in Japan. It sheds light on the psychological reasons behind this behavior and the strength of habituation, which is important for both public health and behavioral science. By addressing some of the aforementioned suggestions and presenting the research questions and objectives more explicitly in the introduction, it would further enhance the clarity and impact of the study. Nonetheless, the study's methodology is robust, and its findings contribute to the understanding of mask-wearing behaviors in Japan, which could be beneficial in the context of public health and behavioral studies.
Author Response
[Responses to the Reviewer #2]
The study under consideration, titled "Habitual Mask Wearing as Part of COVID-19 Control in Japan: An Assessment Using the Self-Report Habit Index," presents valuable insights into the continued practice of mask-wearing in Japan even after the removal of government-mandated requirements in 2023.
- Introduction: The introduction of the article effectively sets the stage by addressing the continued high rates of mask-wearing in Japan, even after official mandates were lifted. However, the introduction could be enhanced by explicitly stating the research questions and objectives of the study. By doing so, the authors could provide a clear roadmap for readers, guiding them on what to expect in the study.
>>
We thank the reviewer for this comment. Study question and purposes were reiterated in P2L76-80.
- Methodology: The methodology section of the study is commendable for its clarity and detail. The use of an Internet-based cross-sectional survey is well-suited to capture a large sample of participants. However, it would be beneficial to further elaborate on the recruitment process and discuss any potential limitations associated with non-random participant recruitment.
>>
Additional description of the sampling procedure was put in Methods (P2L96-P3L98). Potential limitations were discussed in Discussion (P9L301-303)
- Results and Findings: The article reports that only 4.9% of the 2,640 participants surveyed reported not wearing masks at all. It highlights the importance of conformity to social norms as a primary reason for mask-wearing, and it underscores the habituation towards mask-wearing, particularly in public transport. This section presents a clear picture of the study's key findings.
>>
We thank the reviewer for this encouraging comment.
- Discussion: While the discussion section effectively links the findings to the existing literature, it could be strengthened by a more comprehensive exploration of the implications of these findings. For instance, how might this ongoing habituation to mask-wearing impact public health and social norms in Japan? Additionally, the article could discuss potential policy implications and recommendations based on the results.
>>
We agree and we modified Discussion (P9L271-277 and P10L292-294)
- Conclusion: The conclusion succinctly summarizes the key takeaways from the study, emphasizing that mask-wearing in Japan has become a daily habit, particularly in public transport, due to automatic and frequent practice.
- Overall Appreciation: This study is a valuable contribution to understanding the persistent practice of mask-wearing in Japan. It sheds light on the psychological reasons behind this behavior and the strength of habituation, which is important for both public health and behavioral science. By addressing some of the aforementioned suggestions and presenting the research questions and objectives more explicitly in the introduction, it would further enhance the clarity and impact of the study. Nonetheless, the study's methodology is robust, and its findings contribute to the understanding of mask-wearing behaviors in Japan, which could be beneficial in the context of public health and behavioral studies.
>>
We thank the reviewer for these encouraging comments.
Reviewer 3 Report
Comments and Suggestions for Authors.

Author Response
[Responses to the Reviewer #3]
Review comments are attached as pdf file.
- Introduction: In my opinion, the authors should not take sides on either vaccination or PHSMs. Vaccination could have caused so many side effects! PHSMs too. Why take sides?
Row 41 PHSMs are effective? Why?
Row 44 There are many studies that show that the use of masks does not reduce the spread of the virus.
>>
We thank the reviewer for this comment. Following revisions were made:
- We would like to emphasize that we have never intended to take a side with regard to interventions. We agree that it is critical to indicate that the effectiveness of both PHSMs and vaccination are limited. In P1L36-38, we emphasized that both pharmaceutical and non-pharmaceutical interventions have been carried out, and also that none of the interventions were perfect.
- We clearly indicated that PHSMs that restrict contact was shown to have temporarily reduced incidence (P2L47).
- We added a brief memo that individual causal effect of protection by wearing the mask has been yet debated in P2L50-51.
- Methods: The study design and methodology are appropriate to the objectives. The statistical methods appear correct.
- Results: The data presentation and analysis are correct. The results are clear. The graphs and tables are readable and well designed.
- Discussion: The relevance of the results is well summarized in the discussion. It describes the limitations of the study well.
- Conclusions: The conclusions are important and relevant. They relate to the objectives and are based on the results.
>>
We thank the reviewer for these encouraging comments.